# Japanese Encephalitis: Risk of Emergence in the United States and the Resulting Impact

**DOI:** 10.3390/v16010054

**Published:** 2023-12-28

**Authors:** Thomas P. Monath

**Affiliations:** Quigley BioPharma LLC, 114 Water Tower Plaza No. 1042, Leominster, MA 01453, USA; tom@quigleybio.com

**Keywords:** Japanese encephalitis, Flavivirus, *Flaviviridae*, vaccine, veterinary vaccine, emerging disease, zoonosis

## Abstract

Japanese encephalitis virus is a mosquito-borne member of the *Flaviviridae* family. JEV is the leading cause of viral encephalitis in Asia and is characterized by encephalitis, high lethality, and neurological sequelae in survivors. The virus also causes severe disease in swine, which are an amplifying host in the transmission cycle, and in horses. US agricultural authorities have recently recognized the threat to the swine industry and initiated preparedness activities. Other mosquito-borne viruses exotic to the Western Hemisphere have been introduced and established in recent years, including West Nile, Zika, and chikungunya viruses, and JEV has recently invaded continental Australia for the first time. These events amply illustrate the potential threat of JEV to US health security. Susceptible indigenous mosquito vectors, birds, feral and domestic pigs, and possibly bats, constitute the receptive ecological ingredients for the spread of JEV in the US. Fortunately, unlike the other virus invaders mentioned above, an inactivated whole virus JE vaccine (IXIARO^®^) has been approved by the US Food and Drug Administration for human use in advance of a public health emergency, but there is no veterinary vaccine. This paper describes the risks and potential consequences of the introduction of JEV into the US, the need to integrate planning for such an event in public health policy, and the requirement for additional countermeasures, including antiviral drugs and an improved single dose vaccine that elicits durable immunity in both humans and livestock.

## 1. Introduction

Japanese encephalitis is a potentially severe and fatal disease characterized by non-suppurative inflammation and damage to the central nervous system (CNS) caused by a member of the Flavivirus genus, family *Flaviviridae.* JE virus (JEV) is transmitted principally by *Culex* mosquitoes, with pigs and birds as viremic hosts in the amplification cycle. JEV affects a wide area of temperate and tropical parts of the Asia–Pacific region inhabited by over 3 billion people (Figure 1), with an annual incidence of approximately 100,000 human cases and 25,000 deaths [1], and a high proportion of survivors have significant permanent neurological impairment. Notwithstanding preventative vaccination, JE remains the leading cause of viral encephalitis in Asia [2].

Pigs are an important amplifying host in the JEV transmission cycle, which is a major threat to the swine industry as it causes CNS disease, abortion, and reproductive failure. Humans and horses, also affected by the disease, are dead-end hosts and do not develop viremia levels sufficient to infect blood-feeding mosquitoes. There is an extensive literature available on JE history [3], epidemiology [4], vector-host relationships [5,6,7,8], pathogenesis [9,10,11,12], vaccine development, and utilization [13,14,15,16,17].

The focus of this paper is on JE as an emerging viral disease with potential for introduction from Asia and spread in the Americas. This possibility is illustrated by the recent introduction and spread of other mosquito-borne viruses—West Nile (WN), Zika, and chikungunya—from the Old to the New World, the invasion of Europe by another flavivirus, Usutu [18], having a similar transmission cycle as WNV, and the invasion of continental Australia by JEV [19,20,21].

## 2. Historical Threat

Following World War II, there was concern that the return of equipment and material from the Pacific Theatre could lead to the introduction of JEV into the United States, with the establishment of transmission by indigenous mosquito vectors and vertebrate hosts and subsequent geographic spread. This risk was again acknowledged in the 1980s, when the exotic mosquito species and secondary JEV vector [22,23], *Aedes albopictus*, was introduced into the US via ova infesting used truck tires imported from Japan [24]. By that time, vertical transmission of JEV in mosquitoes had been established experimentally [25], and it was feared that JEV could be introduced by naturally infected ova that hatched in used tires left outdoors and filled with rainwater, which served as sites for mosquito oviposition and breeding. After its introduction, *Ae albopictus* spread across the eastern half of the US, from infested to adjacent areas, at a rate of 100–300 km per year [26]. Similarly, another invasive Asian mosquito and secondary vector of JEV, *Ae. japonicus*, which is also capable of transovarial JEV transmission and was first detected in the US in 2000, has likely been repeatedly introduced and has greatly expanded its distribution over 33 states [27].

Although JEV has not been detected in the US, the risk may be reconsidered in light of climate change and changes in the ecology and distribution of JEV and in international travel and trade. A quantitative risk assessment conducted in 2019 considered the potential mechanisms of the introduction of JEV into the US and, based on modeling assumptions, concluded that there was a high risk of introduction by an infected adult mosquito on passenger aircraft during the summer months [28]. Geographically, the risk of introduction into the US may be highest in California due to the frequency of arrivals from Asia and the abundance of *Culex* vectors and avian hosts [29], and the state ranks 10th in the nation in feral pig populations, which could serve as amplifying hosts [30].

The closely related WNV rapidly spread across the US between 1999 and 2003 and also utilizes birds and *Culex* spp. mosquito vectors for transmission; however, swine—a large mammalian amplifying host for JEV—are not involved in WNV transmission. Young pigs are susceptible to neuroinvasion by JEV and clinical encephalitis, whereas, in adult female pigs, JEV infects the developing fetuses and causes abortion and stillbirth. In adult male pigs, testicular infection and swelling may cause infertility. For these reasons, JEV poses a significant threat to the US swine industry.

## 3. The Current Threat

In October 2022, the Swine Health Information Center of the US Department of Agriculture (USDA) Animal and Plant Health Inspection Service (APHIS) held a symposium entitled “Japanese Encephalitis Virus: Emerging Global Threat to Humans & Livestock” and initiated a website a year later promoting preparedness and diagnostic testing, with new goals for veterinary public health coordination, distribution of information, and JEV-specific testing available to the livestock industry [31,32]. Funding has been provided for studies of the pathways of potential JEV introduction into the US. Learnings from the 2022 introduction of JEV into Australia have been incorporated into the analysis of response measures. The level of concern for human health in the US has not been elevated proportionately. Although it may be acknowledged that surveillance for human encephalitis, diagnostic testing, organized mosquito control, and the availability of an FDA-approved JE vaccine represent safeguards that would be available if JEV appeared in the US, it is likely that a major public health and veterinary emergency would occur before those measures resulted in control of the disease.

The risk of expansion of the geographic footprint of JEV is illustrated by events in Australia in the last 2 years. Prior to 2021, JEV (genotype IV) activity had been confined to the tropical islands of the Torres Straits and the Northern Peninsula Area at the peak of the Cape York Peninsula [21]. In early 2021, a human case of JE was diagnosed in a resident of the Tiwi Islands, 80 km north of Darwin in the Northern Territory. A year later, in February 2022, an outbreak of JE was detected in pig farms in southern Queensland, Victoria, New South Wales, and South Australia, followed by multiple human case reports, totaling 42 cases and 7 deaths and constituting the largest virgin soil outbreak of JEV in history. Ardeid birds (herons and egrets) and other birds were implicated as the principal amplifying hosts and *Cx. annulirostris* the principal vector, although other mosquito species may have played a lesser role. It is likely that JEV had circulated and spread southwards undetected in birds, feral pigs, and possibly bat populations for months before being detected in southern Australia, where commercial pig farming is practiced. The Northern Territory and Cape York Peninsula are areas of low domestic pig density, although feral populations are high [21]. This may have reduced the potential for detection of JEV as the virus spread southwards. A program has been initiated to control mosquito breeding in and around piggeries and to vaccinate persons at risk. Modeling indicates that approximately 850,000 Australians reside within the flight range of the principal vector, *Cx. annuloristris*, from a piggery [33]. Both inactivated (JESPECT or IXIARO) and live attenuated (IMOJEV) JE vaccines are licensed in Australia, but a policy for use and defined at-risk target populations for vaccination have not yet been established. There are no approved veterinary vaccines. In short, the appearance of JE in Australia is a dress-rehearsal for emergence of the disease in the US and the short-comings of limited veterinary vaccine availability.

## 4. Mechanisms of Potential Introduction and Spread of JEV

The published quantitative assessment referred to above considered the mechanisms whereby JEV could be introduced [27]. These pathways included (a) infected mosquito vectors (by aircraft, cargo ships, tires, or wind); (b) import of viremic animals or infected animal products; (c) transport by viremic migratory birds; (d) import of infectious or contaminated biological materials (e.g., vaccines); (e) import of infected animal products; and (f) entry of infected humans. The introduction of infected adult mosquitoes was considered the most likely mechanism. However, the source of introduction would probably be extremely difficult to identify and would have occurred weeks or months before recognition, as was likely the case for the WN virus, which was found first in human cases of encephalitis in New York City in 1999. From New York, WNV began a rapid expansion across the entire country over 4 years and became the most common single cause of viral encephalitis in the US [34,35]. In the 20 years since its introduction, there have been an estimated 7 million persons infected with WNV in the US, 51,702 total case reports, 25,227 cases of encephalitis, and 2376 deaths [36]. One aspect of the clinical presentation that might lead to early recognition is the predilection of JE for children [37], whereas WN principally affects adults, with the highest attack rate in the elderly.

Of 41 species that have been implicated in the transmission of JEV by detection in field-collected mosquitoes in the Asia–Pacific region or in experimental studies, a number occur in the US (Figure 2) [5,6,38,39]. The *Cx pipiens* complex (*Cx. pipiens* and *Cx quiquefasciatus*) are competent vectors of JEV, as well as multiple other flaviviruses, including WNV and St. Louis encephalitis virus (SLEV) in the US. It is likely that other North American mosquitoes are competent vectors but have not been evaluated. This question needs to be carefully evaluated to understand the receptivity of the US (and tropical America) to the introduction and spread of JEV.

Like WNV, JEV readily infects birds, which serve as viremic vertebrate hosts and are widely distributed and abundant in the US, including around airports (potential sites of entry), where they represent a hazard to aircraft. The circulation of WNV is often revealed by overt illness and death in birds, particularly corvids, which are highly susceptible, whereas JEV is less pathogenic and clinically silent in avian species. This increases the likelihood that JEV could circulate for some time without recognition, as certainly happened in Australia prior to recognition in 2022. Studies of JEV ecology in Asia have focused attention on wading birds (*Ardeidae*) [40] and domesticated birds (ducks) [41], but the implication of these species in transmission was affected by sampling bias. North American birds, including house sparrows, grackles, starlings, red-wing blackbirds, and rock pigeons, as well as egrets, develop viremia following experimental infection with JEV genotypes I and III [6] and would likely play a role in transmission and spread following an introduction.

In contrast to WN, pigs are highly susceptible to JEV and also serve to amplify JEV transmission by mosquito vectors. In Asia, pigs are believed to be the most important hosts in transmission. The discrepancy in host susceptibility between WN and JE may be mediated by non-structural genes of the virus determining viral replication, as shown for differences between WN and St. Louis encephalitis (SLE) virus infection in avian species [42].

In addition to becoming viremic, experimentally infected pigs also shed JEV virus from nasal epithelium [43], playing a potential role in non-arthropod-borne contact spread [44,45,46]. This route of infection probably plays a role in JEV transmission in crowded swine barn conditions. JEV oral shedding was detected in some experimentally infected North American bird species [6], and mice infected with JEV intranasally shed the virus and can infect other mice by aerosols or direct contact [47]. Of interest, birds also shed WNV orally and in feces and contact spread has been documented experimentally [48].

Shedding of JEV from pigs raises the possibility that the virus could be introduced from Asia by passengers carrying infected pork products, although secondary spread appears to be a very low risk. Illegal introduction of pork products has long been a concern of USDA and US Customs and Border Control for the introduction of African swine fever, swine vesicular disease, and classical swine fever (hog cholera).

Multiple factors in the relationship between hogs, pigs, and JEV underlie the concern regarding introduction of the virus into the US. These include the potential for the spread of JEV by multiple indigenous mosquito vectors, especially *Culex* spp. and the potential for pig–pig contact spread, as well as the reports of persistent infection in pig tonsil and other lymphoid tissues [45]. These factors would likely lead to recommendations for restricted movement of swine and possibly even some depopulation measures, which may have limited effectiveness in the case of a vector-borne disease. Australia has not limited the movement of pigs, pork, or pig semen with the expansion of JEV in the continent and has not recommended depopulation but has focused on reducing mosquito vector populations around piggeries.

There are over 72 million head of domestic swine in the US, concentrated in the Mid-West [49], 60,000 pig farms, and a pork industry that contributes $57 billion to the US economy [50]. Additionally, there are large numbers of feral swine, with over 6 million animals across 35 states, which represent a potential for unrecognized disease transmission, as well as small-scale backyard pig and poultry operations, for which biosecurity measures and veterinary oversight are low [51]. In parts of Asia where they have been studied, feral swine appear to play an important role in JEV transmission and are not subject to preventative immunization [52]. Feral swine populations are expanding in the US, and their distribution is principally in warm climates of the Gulf Coast, from Florida to Texas [53]. This fact favors mosquito-borne transmission. The proximity of feral pigs to airports that might be the points of introduction of infected adult mosquitoes is uncertain, but it is worthy of mention that at least one major international airport in Europe serving Asia has intentionally placed pigs in surrounding fields to prevent bird strikes [54]. JEV represents a threat of introduction into Europe, as it does for the Americas, and there are reports of finding JEV RNA in birds and a pool of *Cx pipiens* mosquitoes sampled in Italy [55,56].

Organized large-scale piggeries in the US are organized with an emphasis on biosecurity measures;, however, these measures principally include precautions against contagious diseases such as pseudorabies, African swine fever, brucellosis, and porcine reproductive and respiratory syndrome. In Australia, as a result of the introduction of JEV, new biosecurity efforts have been revised to include mosquito control activities [57]. These measures have been noted in the USDA’s recent preparedness efforts [30], but implementation, especially for outdoor piggery operations, will be extremely challenging.

## 5. Consequences of and Response to the Introduction of JEV into North America

The introduction of WNV into the US in 1999 was one of the most important events in the modern history of emerging infections prior to the Ebola outbreak in West Africa in 2014–2016 and the global SARS-CoV-2 epidemic in 2019. JE is a much bigger disease threat than WN to human and animal health in the US. There is no barrier to spread by indigenous mosquito species and vertebrate hosts other than the background of cross-protective immunity to the antigenically related WNV [58]. Cross-protection has been demonstrated experimentally in rodents [59], nonhuman primates [60], and wild birds [61]. Cross-protective immunity to WNV could dampen transmission of JEV in avian hosts, as was postulated for the displacement of SLEV in southern California [62]. Large-scale indoor pig operations practicing high-level biosecurity measures represent a barrier to JEV transmission, and mosquito control measures, as now recommended in Australia, would likely be introduced.

Introduction of JEV would constitute a public health emergency that would require a substantial response and, if transmission was established, would constitute a blow to the economy. It is likely that an initial focus of transmission would go unnoticed, and that the virus would become established before mosquito-control measures could be taken to eradicate it. The predilection of JEV for children, the high case–fatality rate, and the difficulty of preventing mosquito exposure in children engaged in outdoor activities would create significant concerns.

Following a point introduction of JEV into the US, rapid spread would be expected, both radially and over long distances by mobile and migratory movements of birds [63]. Bats are also involved in JE transmission in Asia [64] and represent another mechanism for geographic spread. The introduction of JE into the US would evoke a substantial effort on surveillance of birds, pigs, and mosquitoes on the part of local, state, and federal public health agencies, including adding JEV diagnostic test methods to nationwide clinical laboratory services and intensified vector control activities. Local laboratory-based surveillance for mosquito-borne diseases had been actively practiced in the US through the 1980s, but cost factors and competing priorities led to the senescence of many programs; these activities were temporarily re-stimulated by the WN outbreak, and nationwide reporting of arbovirus activity remains in place [65,66]. Surveillance and monitoring of JEV in birds and swine, and human and equid case diagnosis by serological methods, are complicated by the immunological cross reactivity between JE, WN, and SLE viruses.

## 6. Vaccines as a Countermeasure

Although there is considerable genetic variation of JEV strains, and all marketed vaccines are derived from a single JEV genotype (Genotype III), human vaccines are believed to protect against all strains in the five known virus genotypes. However, in veterinary practice in Asia, there has been some concern about lower efficacy of JEV Genotype III vaccines against the dominant circulating Genotype I strains [67], and, in consequence, a Genotype I live vaccine is in development for use in swine in South Korea. Similar concerns for reduced immunogenicity of Genotype III based vaccines against the highly pathogenic Genotype V have also been raised. A recombinant live vaccine based on Genotype I, with the envelope sequence of Genotype V and the attenuating mutations found in the SA14-14-2 attenuated vaccine, is in preclinical studies [68].

There are four types of human JE vaccines approved for use in Asia, all based on Genotype III: two inactivated whole virion vaccines, produced in mouse brain tissue and in Vero cell culture, and two live, attenuated vaccines—SA14-14-2 produced in primary hamster kidney cells and a recombinant (chimeric) yellow fever 17D-JE (SA14-14-2) vaccine produced in Vero cells. Given available alternatives, the animal tissue vaccine substrates (mouse brain and hamster kidney) would not be acceptable for use in the US today. A recent review indicated that the inactivated Vero cell and live, attenuated recombinant (chimeric) JE vaccines were safe and highly immunogenic [15] and are further discussed below. In the US, only one vaccine, the Vero cell inactivated JE vaccine (IXIARO^®^), is approved by the FDA for use in persons 2 months of age or older.

There is a long history of use of JE vaccines for the immunization of pigs in Japan, Nepal, Taiwan, and the Republic of Korea, principally aimed at the protection of pregnant sows against abortion, stillbirth, hydrocephalus, and other congenital malformations. Other countries, including Malaysia, Thailand, Sri Lanka, and China, have limited swine immunization programs. Effectiveness of the vaccination of pregnant sows has been shown both experimentally and in field studies [69]. Both inactivated and live attenuated vaccines have been employed, the latter appearing to be most effective [70]. There is a large body of research on newer vaccine candidates, including recombinant protein [71], plasmid DNA [72], lentiviral vector [73], virus-like particle [74], mRNA lipid nanoparticle vaccine [75], and recombinant live vaccines based on the Genotype I backbone [67,76]. It is unlikely that any of the veterinary vaccines currently used in the endemic region would be acceptable for importation and use in the US in an emergency without considerable additional study, and the newer candidates are early in development. It may be concluded that the existing vaccines approved for humans should be explored as veterinary vaccines.

Immunization of pigs as a public health measure to minimize virus amplification and prevent human disease has not been demonstrated. This goal is complicated by (i) the need to rapidly protect gilts born during the summer months of virus transmission and (ii) by the immunological immaturity of piglets during their first month of life. Passive transfer of immunity is inefficient; colostrum and milk from immune sows prevents JEV infection in gilts for only a short period, since the ability to adsorb immunoglobulins from the gut is limited after 4 weeks of age and since the porcine placenta does not permit the transfer of maternal antibody from the bloodstream [77].

If JEV were introduced into the US swine population, it would likely take up to 2 years for a veterinary vaccine to be conditionally approved. A high priority would be for the development of a safe, live attenuated vaccine that elicited rapid protection without the need for booster doses. Having this preventive measure stockpiled and available in an emergency would likely be a priority for US homeland security and would avoid the scramble to produce such an important health measure after the fact. This point is illustrated by the introduction and spread of WNV, which caused deaths and required euthanasia of many horses and valuable zoo animals in the US before a veterinary vaccine was approved, and by the widespread disease in Australian piggeries in the ongoing JEV outbreak. Until a veterinary JE vaccine is approved and commercialized, the focus in the US would be on fire-fighting outbreaks with vector control around piggeries and by limiting the movement of pigs.

Neutralizing antibodies constitute a surrogate for JE vaccine efficacy [78,79], and a neutralization titer of ≥10 has been accepted as a correlate of protection by the World Health Organization [80] and ACIP [72]. Only a low concentration of antibody is required to prevent neuroinvasion by the virus. IXIARO was therefore approved by the FDA for human use based on neutralizing antibody as a correlate of clinical benefit, and no post-marketing requirements were imposed for demonstrations of efficacy in reducing disease [81]. The ACIP provided recommendations for use of the inactivated JE vaccine (IXIARO) for adult US travelers and laboratory workers in 2010 [82] and for children in 2013 [83]. Whereas human vaccination against JE has been cost-effective in JE endemic countries with high burdens of disease [84], the history of immunization of travelers in the US, Australia, France, and other countries suggests that vaccine uptake is relatively low and the cost per case averted is very high [85,86,87]. This equation would likely change if the virus were introduced into the homeland, but, as yet, no strategy for such an event and no policy on the use of human vaccination if JEV were introduced has been set forth by CDC [88].

IXIARO has been studied in multiple clinical trials and has been shown to be safe and immunogenic [70]. The vaccine is produced by formalin inactivation of purified SA14-14-2 JE virus from Vero cell culture fluid. The SA14-14-2 strain is an attenuated virus, which has been used as a live vaccine in China and some other Asian countries since 1989 [89]. Although two doses are required for primary immunization with IXIARO, these may be administered at a short interval (7 days) in young adults, although a 28-day interval is recommended for persons ≤18 or ≥65 years of age [90]. Protective immunity is assumed to be established within 1 week after the second dose. Antibody titers wane over time, and boosting to maintain immunity is recommended within 11 months of primary immunization. IXIARO is marketed globally by Valneva, the parent company, and by multiple distribution partnerships in Australia, Europe, and Asia. The utility of IXIARO for immunization of livestock has not been determined.

Whereas the profile of IXIARO is certainly acceptable in the event risk-based vaccination was recommended in the US, there would be advantages for a vaccine that elicits rapid protection after a single dose in all age groups and that provides long-term immunity (at least 5 years) without the need for boosting. A live, attenuated single-dose vaccine with this product profile, IMOJEV^®^, was developed in the US as a chimeric virus in which the envelope genes of JEV (the SA14-14-2 strain) are inserted into the backbone of yellow fever 17D vaccine virus, a live vaccine with a long history of use [91]. IMOJEV is manufactured in Vero cell cultures to international standards, is marketed in Australia and in a number of Asian countries (Brunei, Cambodia, Hong Kong, Indonesia, Laos, Malaysia, Myanmar, Philippines, Singapore, Thailand, and Vietnam), and has been widely used with an excellent safety and immunogenicity record [16,70,92]. Since it is a live vaccine, it is contraindicated in pregnancy and for individuals with immune deficiency disorders. The vaccine was marketed by Sanofi Pasteur until 2022, when it was acquired by Substipharm Biologics [93]. It is not approved in the US or Europe but would be a useful addition to preparedness efforts in the event of emergence of JE there. The utility of IMOJEV for immunization of livestock has not been determined and deserves study.

In addition to vaccines, antiviral drugs could play an important role in the treatment of JE and other flavivirus infections. Although there are some promising approaches [94], none are in clinical development.

## 7. Conclusions

Japanese encephalitis poses a material threat of introduction and spread in the US and tropical America, and there is a need to consider steps to prepare for and mitigate this eventuality. As a result of the introduction of JEV into continental Australia in 2022, the USDA has initiated measures aimed at preparedness in the US because of the danger JEV poses to the swine industry and the economy as a whole. With respect to human health, no threat assessment or plan has yet been made public by the CDC. Fortunately, there is an FDA-approved JE vaccine for use in children and adults. Policies for vaccination have yet to be promulgated and will be required based on the geographic impact, incidence, and risk factors for JE infection, and supply constraints may be an issue since the current indication is only for travelers, laboratory workers, and military personnel. Consequently, consideration should be given to an emergency use stockpile to ensure immediate availability in the event of an outbreak and to promoting the approval of the live, attenuated single-dose vaccine, IMOJEV, that elicits durable immunity. Other preparatory public health measures may include making available rapid, specific diagnostic tests and surveillance activities, particularly around potential sites of introduction. Further research is also needed on the competence of indigenous mosquito vectors, mechanisms of overwintering of the virus in the US, the potential for direct contact transmission in animals, the interaction of JEV and other flaviviruses in vectors and hosts, and the development of effective antiviral drugs.

## Figures and Tables

**Figure 1 viruses-16-00054-f001:**
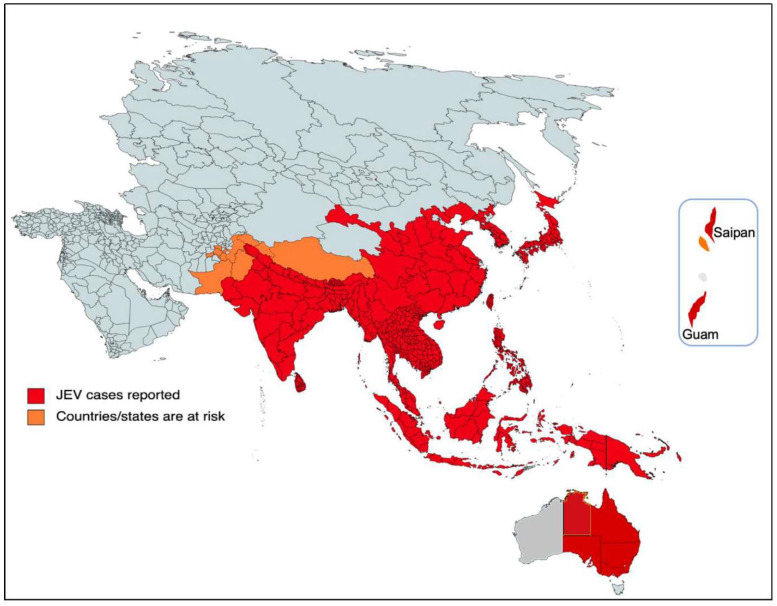
Geographic distribution of Japanese encephalitis in the Asia–Pacific region. The possible occurrence of JEV in birds and in a Culex pipiens mosquito pool in Italy is not shown. From Scholarly Community Encyclopedia (encyclopedia.pub/entry/43099, accessed on 5 December 2023).

**Figure 2 viruses-16-00054-f002:**
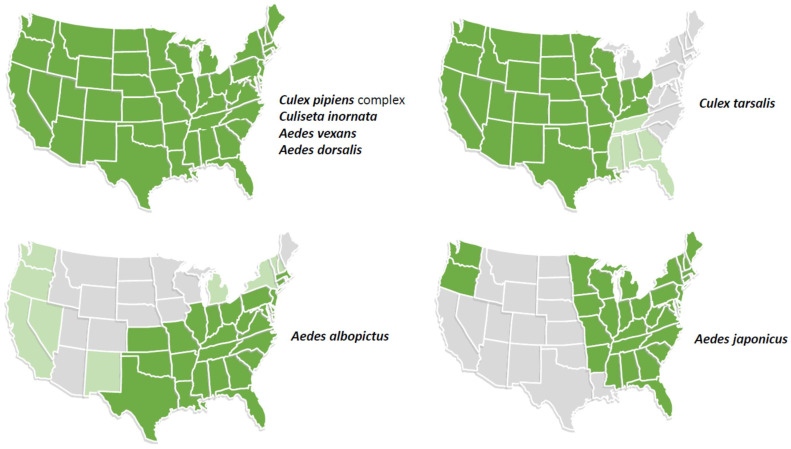
Distribution in the continental United States of mosquito species that are known vectors of JEV based on detection of virus in mosquito pools in the Asia–Pacific region or that are potential vectors based on experimental studies. *Aedes albopictus* and *Ae. japonicus* are invasive species. Light green shading indicates states with low prevalence or incomplete distribution of the vector.

## Data Availability

Data are in published sources cited in the paper.

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
