# Peer review of "Japanese Encephalitis: Risk of Emergence in the United States and the Resulting Impact"

_viruses, 2023, doi:10.3390/v16010054_

Round 1
Reviewer 1 Report
Comments and Suggestions for Authors
The review by Monath is really timely with increasing JEV risk across the world. With the outbreak in Australia in early 2022, there is now a real risk of introducing the virus in the US. Apart from the public health issues, there will be a threat to the pig industry if JEV was to cause outbreaks. The review covers this aspect of the JEV risk to the US.
One suggestion is inclusion of some details of the Australian outbreak with the genotype IV, which seems to be on the rise in Asia.
Author Response
Thanks to the reviewer for his/her suggestions.
In response, the section of the paper dealing with the Australian epidemic has been modified and expanded (lines 85-106)
Reviewer 2 Report
Comments and Suggestions for Authors
The author reviewed the potential threat of JEV to US. Although current threat of JEV in US is well =described, scientific evidence that live recombinant JE vaccine can protect pigs. Comments for the authors below:
Major points:
1. Lines 216-217: Please include more references containing recombinant vaccine efficacy in animals as below.
a. S. Tajima, S. Taniguchi, E. Nakayama, T. Maeki, T. Inagaki, M. Saijo, C.K. Lim, Immunogenicity and Protective Ability of Genotype I-Based Recombinant Japanese Encephalitis Virus (JEV) with Attenuation Mutations in E Protein against Genotype V JEV. Vaccines (Basel), 2021 Sep 25;9(10):1077. doi: 10.3390/vaccines9101077.
------------------------------
- What is the main question addressed by the research?
The author reviewed the potential threat of JEV to US.
- Do you consider the topic original or relevant in the field? Does it
address a specific gap in the field?
Yes, but not enough.
- What does it add to the subject area compared with other published material?
The author is recommending to consider the use of recombinant live viral JEV vaccines.
- What specific improvements should the authors consider regarding the methodology? What further controls should be considered?
Reviews of the vaccine efficacy in pigs (or animals) are missing.
- Are the conclusions consistent with the evidence and arguments presented and do they address the main question posed?
Evidence of the vaccine efficacy in pigs (or animals) are missing.
- Are the references appropriate?
Require more references concerning vaccine efficacy in veterinary animals.
- Please include any additional comments on the tables and figures.
None.
Author Response
Response to reviewer #1
Thank you for your comments and suggestions.
Major point 1. Line 216-217
The point being made in this section of the paper is that there was no vaccine available when WNV was introduced into the US in 1999 and there is no veterinary vaccine against JEV approved currently in the US. This creates a vulnerability that could be addressed by efforts to develop and gain regulatory approval of a veterinary JE vaccine. Only one human vaccine is approved in the US.
Reviewer’s points 1,2,4,5, and 6 focus on the need to add additional information on veterinary vaccines against JE.
Additional information on veterinary vaccine types, availability, utilization of vaccines in Asia, including veterinary vaccines, as well as information on new approaches to JE vaccines has been added, inserting several paragraphs and a number of references (lines 201-224 in the revised manuscript). This section now includes reference to the Tajima et al paper cited by the reviewer.
I do wish to point out that this paper is not intended to be an exhaustive review of the pros and cons of JE vaccine technologies. The aim of the paper is to raise awareness of the potential for JE emergence in the US, recently heightened by the events in continental Australia, and the approaches that might be considered with respect to preparedness for such an emergency. Vaccines are an important component of threat reduction.
With regard to Question 3 (how does the subject area compare to other published material?) the reviewer comments that the author is recommending live attenuated vaccine. In fact, the paper aims to address the new (or heighted) threat of emergence of JE in the New World. Introduction of a live vaccine licensed outside of the US but meeting US-standards for vaccine quality is suggested as one measure that could improve preparedness. There are relatively few publications on the topic of JE emergence in the US, and these are cited in the paper. The author believes that the subject is timely and the paper adds considerably to the available literature. The recent USDA conference of the threat of JE introduction is not published, and is available only by listening to lengthy oral presentations on You-Tube.
Reviewer 3 Report
Comments and Suggestions for Authors
This study reviews the latest reports on the route of Japanese encephalitis virus entry into the United States, the spread of infection, and the effectiveness of vaccines against it. It is worthy of publication in this journal because it clarifies the position of the conclusions of the latest symposium and newly developed vaccines while referring to past findings. However, regarding the following points, you should appropriately refer to the papers you have already referred to.
1. The authors report on the introduction of Japanese encephalitis virus into the United States by the Symposium of Swine Health Information Center of the US Department of Agriculture (USDA) Animal and Plant Health Inspection Service (APHIS) entitled “Japanese Encephalitis Virus: Emerging Global Threat to Humans & Although the year of invasion is estimated by referring only to "Livestock", The authors have already referred to Oliveira, Ana RS, et al. "Introduction of the Japanese encephalitis virus (JEV) in the United States–A qualitative risk assessment." While further referring to the contents of Transboundary and emerging diseases 66.4 (2019): 1558-1574, it should be indicated that there are opinions that differ from those of the Symposium.
2. JEV and WNV vectors and their distribution should be included in the discussion, with reference to the following literature: Tolsá-García, María José, et al. "Worldwide transmission and infection risk of mosquito vectors of West Nile, St. Louis encephalitis, Usutu and Japanese encephalitis viruses: A systematic review." Scientific Reports 13.1 (2023): 308
3. Considering the introduction into the United States, the authors should refer to the paper below about the Japanese encephalitis virus outbreak in Australia and add a discussion about the route of entry from Australia to the United States and the reliability of the vector.
Hoad, Veronica C., et al. "An Outbreak of Japanese Encephalitis Virus in Australia; What Is the Risk to Blood Safety?." Viruses 14.9 (2022): 1935.
Mackenzie, John S., et al. "Japanese encephalitis virus : The emergence of genotype IV in Australia and its potential endemicity." Viruses 14.11 (2022): 2480.
Author Response
Response to Reviewer #2
The author thanks the reviewer for his/her comments.
Comment 1. “ 1. The authors report on the introduction of Japanese encephalitis virus into the United States by the Symposium of Swine Health Information Center of the US Department of Agriculture (USDA) Animal and Plant Health Inspection Service (APHIS) entitled “Japanese Encephalitis Virus: Emerging Global Threat to Humans & Although the year of invasion is estimated by referring only to "Livestock", The authors have already referred to Oliveira, Ana RS, et al. "Introduction of the Japanese encephalitis virus (JEV) in the United States–A qualitative risk assessment." While further referring to the contents of Transboundary and emerging diseases 66.4 (2019): 1558-1574, it should be indicated that there are opinions that differ from those of the Symposium”
Response: The USDA Symposium included a paper presented by Natalia Cernicchario N. "A model of JEV Importation risk" , and she is a member of the group that published the referenced models on mechanisms of JEV introduction (Oliveira ARS, Piaggio J, Cohnstaedt LW, McVey DS, Cernicchiaro N. Introduction of the Japanese encephalitis virus (JEV) in the United States - A qualitative risk assessment. Transbound Emerg Dis. 2019 Jul;66(4):1558-1574.) At the USDA Symposium, Cernicchario presented virtually the same material and conclusions reported in the 2019 publication. The author could not find a contradictory opinion and would be grateful for clarification on that point.
Comment 2. JEV and WNV vectors and their distribution should be included in the discussion, with reference to the following literature: Tolsá-García, María José, et al. "Worldwide transmission and infection risk of mosquito vectors of West Nile, St. Louis encephalitis, Usutu and Japanese encephalitis viruses: A systematic review." Scientific Reports 13.1 (2023): 308
Response: This reference has been added (ref 40); it is indeed a comprehensive review, but does not change the conclusions in the paper regarding the potential of North American mosquito species to be infected with or transmit JEV (Fig 2). The vector capacity of mosquitoes for WNV is beyond the subject matter of this paper, and it is not possible to speculate that WNV vectors such as Cx. restuans would play a role in JEV transmission without experimental evidence. The fact that the Cx. pipiens complex species are vectors for both WN and JE viruses has been added in the brief paragraph on mosquito vectors (lines 108-111).
Comment 3. Considering the introduction into the United States, the authors should refer to the paper below about the Japanese encephalitis virus outbreak in Australia and add a discussion about the route of entry from Australia to the United States and the reliability of the vector.
Hoad, Veronica C., et al. "An Outbreak of Japanese Encephalitis Virus in Australia; What Is the Risk to Blood Safety?." Viruses 14.9 (2022): 1935.
Mackenzie, John S., et al. "Japanese encephalitis virus : The emergence of genotype IV in Australia and its potential endemicity." Viruses 14.11 (2022): 2480.
Response: The references have been added.
Round 2
Reviewer 3 Report
Comments and Suggestions for Authors
Since the authors have provided sufficient and appropriate responses and improvements to the Referee's comments, I judge that this paper should be accepted as is.